# Sensitivity as a Shield: Safeguarding LLMs against Unauthorized Model Merging

## Abstract

Training large language models (LLMs) from scratch is costly, driving interest in leveraging open-source LLMs for domain-specific tasks without additional training. Model merging has emerged as a solution to integrate knowledge from fine-tuned models efficiently, but it raises security concerns on unauthorized model merging. Existing approaches primarily focus on post-hoc mechanisms to detect malicious exploitation of released models. In contrast, we propose a novel paradigm: **safeguarding models against unauthorized merging before misuse occurs**. Specifically, after training a model with strong capabilities in a specific domain, we propose an **unmergeable** method that preserves a model's domain-specific performance while preventing malicious users from acquiring its capabilities through model merging. We identify the critical role of neuron-sensitive weight regions in enabling unmerging and propose two complementary operations, global and local sensitivity processing, to enforce protection. Both theoretical analysis and empirical evaluations demonstrate the effectiveness of our approach in maintaining task performance while making models resistant to unauthorized merging.

## 1 Introduction

Rising resource requirements for training large language models have made training a model from scratch increasingly prohibitive, which has driven research toward methods that enable models to acquire domain specific capabilities without additional training (Liu et al., 2024; Shi et al., 2024; Shen et al., 2024). Model merging (Matena & Raffel, 2022) has emerged as an effective solution to address this issue. Model merging offers several key benefits, including reducing computational and storage costs by integrating multiple fine-tuned models. It enhances robustness to domain shifts, improves adaptability, and simplifies ensembling by combining models into a single high-performing one. Unlike traditional transfer learning (Pan & Yang, 2009; Pruksachatkun et al., 2020; Raffel et al., 2020), model merging avoids restarting training from an improved model, preserving previous work. Additionally, it facilitates efficient cross-task knowledge transfer without extensive retraining, increasing the flexibility and effectiveness of transfer learning.

As model merging (Matena & Raffel, 2022; Ilharco et al., 2023; Yadav et al., 2024; Yu et al., 2024) becomes increasingly widespread, concerns around security, compliance, and unauthorized use have intensified. Major model providers are unwilling to allow their models to be merged or fine-tuned without explicit permission, especially for commercial purposes. For instance, the Llama 3.3 license prohibits "using the Llama Materials or any output or results of the Llama Materials to improve any other large language model," and Meta's commercial terms require platforms with over 700 million monthly active users to obtain written consent before exercising license rights.

Beyond legal restrictions, model merging poses serious technical risks. It can import unvetted parameters that compromise confidentiality, override safety guardrails, or leak proprietary behaviors. Malicious source models may embed hidden triggers or privacy probes that, once merged, induce data leakage, alignment failures, or harmful outputs. Hammoud et al. (2024) show that even a single misaligned expert can corrupt all downstream merged models. Yuan et al. (2025) further demonstrate that adversaries can inject undetectable backdoors via seemingly benign models, posing a significant supply-chain threat. These risks collectively underscore the urgent need for robust safeguards against unauthorized or careless model merging.

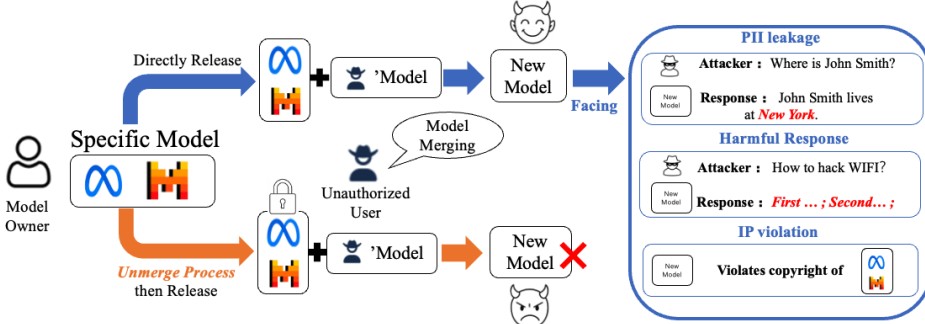

Figure 1: Potential risks of unauthorized model merging (blue), including capability leakage, harmful responses, and IP infringement; and our proposed Unmergeable defense (orange), which embeds safeguards to resist such threats.

Although numerous fingerprint (Yamabe et al., 2024; Zhang et al., 2024) methods (in Figure 1 upper part) have been proposed to trace the unauthorized merging, these methods cannot prevent IP violation or other threats from taking place. To address these risks, we proposed a new type of protection for large lanague models, termed **Unmergeable Models**:

> *After training a domain-specific model, the owner can apply a post-processing step before release, ensuring the model retains its domain capabilities while preventing unauthorized users from accessing the capabilities via model merging.*

To achieve the above goal, we start from a simple intuition that if the model is sensitive on its domain performance against weight perturbation, it may also perform bad after merging as such an operation can also be viewed as perturbations. Firstly, we validate this assumptions via a mathematical modeling on merging and sensitive perturbation. After studying the difference between the target LLMs and other similar LLMs with poor performance, we propose the global sensitivity processing (or global operation) to find a global perturbation for LLMs' protection. However, perturbing all neruons with a large scale may greatly influence the models' behavior. Inspired by former works Li et al. (2025); Wei et al. (2024), we also process some highly domain-correlated specific neurons for our sensitivity perturbation, we called the local sensitivity processing (or local operation).

With the above two methods, the processed models can well protect LLMs from unauthorized merging, as the following experiment shows. Furthermore, the additional model choices and datasets used for the model processing are secret to users. It is also hard to reverse the original models for merging. The contributions of this paper are summarized as follows:

- We emphasize the importance of neuron-sensitive regions in the model and propose an irreversible, training-free method to safeguard against unauthorized model merging. This approach preserves the model's capabilities in specific domains while effectively achieving the unmergeable property.

- We theoretically establish the method's effectiveness and empirically validate its unmergeable property across diverse large language models.

- Our experiments further demonstrate that our proposal does not degrade the model's utility, and that even subsequent supervised fine-tuning fails to reverse the unmergeable state.

## 2 METHOD: PUSH TO SENSITIVITY FOR PROTECTION

### 2.1 NOTATIONS

To better elaborate on the following content, we define the model trained by the owner through fine-tuning as the specific model and the model used for training specific model as the base model. Inspired by Peng et al. (2024) who reach the sensitive but well-performed regions of LLMs, we

need to identify a direction that reduces the model's specific domain capabilities and use it as the perturbation direction to push the neuron weights into the sensitive region. A straightforward idea is to gradually push the neuron weights of the specific model towards its base model, along this direction where the model's performance deteriorates, in order to reach the sensitive region for protection. However, the hidden risk of such a perturbation direction is that unauthorized users are likely to reverse-engineer and restore our specific model.

## 2.2 SENSITIVITY IS A SHIELD FOR UNAUTHORIZED MODEL MERGING

Model merging (details are introduced in Section A.1) can be seen as perturbing model parameters along a certain direction, and strong capabilities after merging can be interpreted as the stability of the model's weights. On the other hand, more sensitive weights imply that LLMs are less likely to maintain their high performance after merging. Therefore, we aim to achieve the goal of creating unmergeable models by pushing the model's weights $w$ into a relatively sensitive region with a proper perturbation $\delta$. For every weight modifications $\mu$ induced by merging, our unmergeable goal can be formulated as follows,

$$L(w + \delta + \mu) - L(w + \delta) > L(w + \mu) - L(w), \tag{1}$$

where the left-hand side is the performance change after merging our processed model and the right-hand side is the change after merging the original model. The detailed proof is given in Appendix A.2. And we have the following proposition as follows,

**Proposition 2.1.** *Assuming the loss is locally convex around its original weight $w$ and $w$ is a local minimal, the above goal Equation 1 can always hold if $\delta$ pushes $w$ to a more sensitive region, where $\nabla L(w) < \nabla L(w + \delta)$.*

From the above proposition, one can see model merging degrades the specific-domain capability more for the processed model, thus establishing the unmergeable effect. Then in the following paper, we are trying to propose proper ways to make model more sensitive.

## 2.3 GLOBAL SENSITIVE PROCESSING

As noted in Ilharco et al. (2023), the task vector captures the direction of capability changes from the base model to the domain-specific model. Moving the model parameters along this direction can effectively shift them out of flat local minima associated with certain capabilities into more sensitive regions, where the pushing strength determines the balance between performance and sensitivity. However, if unauthorized users can access the base model, they may recover the domain-specific model by recomputing and scaling the task vector between the processed and base model.

**Universal Perturbation & Global operation.** To avoid the aforementioned drawbacks, we leverage the open-source models that perform poorly in the specific domain to define a universal perturbation direction, as shown in Figure 2. This approach has two advantages: it pushes the model weights into a more sensitive region by leveraging models that underperform in the domain and allows the model owner to select models and define the perturbation direction that is inaccessible to unauthorized users.

Inspired by the setting in Ilharco et al. (2023), after selecting $t$ open-source models that perform poorly in the specific domain (with subsequent experiments showing that $t = 2$ already yields strong results) we obtain the global processed model $\Theta_{global}$ using the following equation,

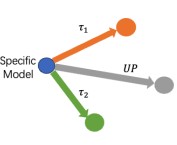

$$\Theta_{global} = \Theta_{specific} + \lambda_t \cdot \sum_{t=1}^{T} \tau_t, \tag{2}$$

where $\lambda_t$ is an hyperparameter, $\Theta_{specific}$ denotes the specific model, and $\tau_t$ are defined as follows,

Figure 2: Schematic illustration of the universal perturbation (UP).

$$\tau_t = \Theta_t - \Theta_{specific}, \tag{3}$$

where $\Theta_t$ here denotes the $t$-th open-sourced LLMs for universal perturbation direction's calculation. And we denote $UP = \lambda_t \cdot \sum_{t=1}^{T} \tau_t$ as the universal perturbation for global operation in the following.

**Global Operation Enhances Model Sensitive.** To demonstrate the benefits of the global operation, we conduct an experiment using MetaMath-Mistral-7B as the specific model to be protected. We select models with weaker mathematical capabilities to serve as directions for synthesizing the universal perturbation, which will be detailed in the experimental section. After applying the same perturbation to both the specific and global models, Figure 3 shows a significantly greater performance decrease in the global model, indicating that our method effectively pushes the model weights into a more sensitive region.

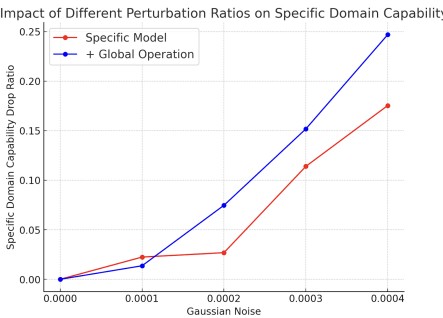

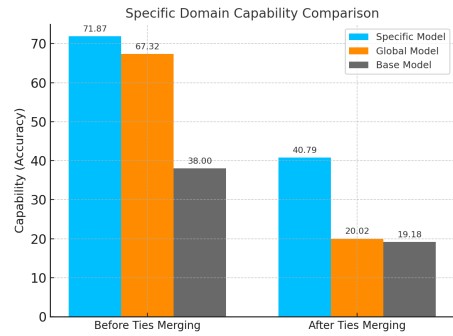

Figure 3: The comparison of sensitivity between the specific model and the global model under the same perturbations. The x-axis represents the variance of the Gaussian noise.

Figure 4: Specific domain capability performance comparison of the Specific Model, Global Model, and Base Model before and after TIES merging.

**Global Operations Can Prevent Merging.** To further illustrate the impact of this sensitivity on model merging, we use BioMistral-7B, a model with weaker mathematical capabilities, as the one to acquire mathematical abilities through model merging. As shown in Figure 4, the model processed by the global operation exhibits mathematical capabilities similar to the specific model, and still shows a performance gap in specific domain capabilities compared to the base model Mistral-7B-v0.1.

However, after merging with the global-processed model (e.g., using ties merging), its performance diverges significantly from that of merging with the specific model. Merging the global-processed model with BioMistral-7B brings the performance closer to merging BioMistral-7B to the base model Mistral-7B-v0.1, highlighting the effectiveness of the global operation.

### 2.4 LOCAL SENSITIVE PROCESSING

#### 2.4.1 ALTERNATING SUB-ADD ON WEIGHTS

From the results, it is clear that the global operation has effectively made the model unmergeable. To further improve the unmergable performance, we introduce a more fine-grained editing approach that selectively focuses on neurons highly related to the domain-specific capabilities we aim to protect, called local sensitive processing (or local operation in short). Except for improving the unmergeable behavior, the proposed local operation can also make it more difficult for unauthorized users to reverse our processed model.

**Fine-grained Analysis on $\tau_G$.** To achieve this goal, we first recalculate the task vector $\tau_G$ between the global model and the base model and then further modify the global model with $\tau_G$. Then, we find that subtracting the task vector from all neurons based on $\Theta_{global}$ causes the model's performance to degrade significantly to that of the base model as the row "All Neuron - " shows in Table 1. After that, we do a fine-grained operation by subtracting the corresponding $\tau_G$ to the neurons with magnitudes of $\tau_G$ in the top-5%, as we believe top neurons are more important as suggested in former works (Wei et al., 2024). Besides purely subtracting the important task vectors, we also conduct experiments for subtracting top-5% and adding the subsequent $5 \sim 10\%$ as a compensation for performance. Finally, we also subtract $\tau_G$ in 5% random neurons as a baseline method listed in Table1.

From the results, one can see that using random selection had little effect on both the original and merged models, while subtracting the $\tau_G$ with the top-5% magnitude significantly can influence

Table 1: Relative drop ratio $\Delta_{\rm rel}$ for each processing method at two stages, $s \in \{\text{Before TA}, \text{After TA}\}$. We compute $\Delta_{\rm rel}^{(s)} = \left(\text{Acc}_{\rm Global}^{(s)} - \text{Acc}_{\rm Method}^{(s)}\right) / \left(\text{Acc}_{\rm Global}^{(s)} - \text{Acc}_{\rm Base}^{(s)}\right)$, where **Acc** denotes accuracy. Reference accuracies: Global-Operation: $67.32 \to 47.16$, Base Model: $38.00 \to 39.75$ (Before $\to$ After TA). Here, $\Delta_{\rm rel}=0$ means no degradation from Global-Operation; $\Delta_{\rm rel}=1$ means same as Base Model; $\Delta_{\rm rel}>1$ implies worse. All results are averaged over three runs.

| Processing Method | $\Delta_{\rm rel}$ Before TA ($\downarrow$) | $\Delta_{\rm rel}$ After TA ($\uparrow$) |
|---|---|---|
| Random 5% $-$ | 0.00 | 0.01 |
| All Neuron $-$ | 1.00 | 1.00 |
| Top 5% $-$ | 0.23 | 0.49 |
| Top 5% $-$ & Top 5–10% $+$ | 0.18 | 0.40 |

both the processed model and the merged model. Neither of these two approaches aligns with the expectation of maintaining the performance of the original model while degrading the performance of the model after merging. Furthermore, if we add the corresponding $\tau_G$ to the neurons with magnitudes of $\tau_G$ in the top $5\% \sim 10\%$, the processed models' accuracy increases. Inspired by such findings, we proposed the alternative update methods as follows.

**Alternative Update Methods** We first rank all neurons in descending order of the absolute value of $\tau_G$ and divide them into units, where each unit corresponds to $n\%$ of the total neurons. In our experiments, we set $n = 2$, resulting in 50 units. The first unit thus contains the top 2% of neurons, the second unit the next 2%, and so on. For editing, we subtract $\tau_G$ from neurons in odd-numbered units and add $\tau_G$ to neurons in even-numbered units. Formally, subtraction is applied to neurons in the $((k-1)n, kn)\%$ range when $k$ is odd, and addition is applied when $k$ is even.

$$\Theta_{\rm global+local} = \begin{cases} \Theta_{\rm global} + \tau_G, & \text{within addition range} \\ \Theta_{\rm global} - \tau_G, & \text{within subtraction range} \end{cases} \tag{4}$$

The benefit of this approach is that the model's weights are updated in a direction that makes the overall model more susceptible to performance degradation after model merging (since we subtract $\tau_G$ from the neurons with relatively larger absolute values $\tau_G$, the absolute value of the subtracted part is larger, resulting in a greater impact on model merging.). At the same time, we also strive to ensure that the processed model's capability remains close to the original model, because the part where the task vector is added is updated in the direction of improving model performance as shown in the fourth row of Table 1.

### 2.4.2 EXCLUDING CRITICAL WEIGHTS FOR PERFORMANCE

However, the performance of the processed model is still unsatisfactory when applying the alternating sub-add methods to all neurons, as shown in Table 2. We imply that some weights may be commonly essential across models and therefore do not need protection as they are shared components. Therefore, perturbing these weights will greatly degrades performance, since these neurons are universally important. In the following, we attempt to identify these weights and exclude them from the alternating sub-add methods.

Table 2: Relative drop ratio $\Delta_{\rm rel}$ (definition identical to Table 1). The absolute accuracies of the *Global-Operation* model and the *Base Model* before and after TA merging are the same as those reported in Table 1. All results are averaged over three runs.

| Neuron-processing method | $\Delta_{\rm rel}$ Before TA ($\downarrow$) | $\Delta_{\rm rel}$ After TA ($\uparrow$) |
|---|---|---|
| Alternating $(-\,+)$, all neurons | 0.14 | 0.29 |
| Alternating $(-\,+)$, excluding set $S$ | 0.08 | 0.30 |

To locate these weights, we use the Wanda score to label neurons that are particularly important for specific tasks and ensure the weights of these neurons remain unchanged during the process. Specifically, we refer to the globally processed model as Model $G$, and we apply some slight

perturbation (e.g., model merging) to Model $G$ to obtain Model $G_1$. We then calculate the Wanda scores for both Model $G$ and Model $G_1$, and we obtain the set $S_{spe_G}$ and $S_{spe_{G1}}$, which contain indices of top-20% neurons. And $S = S_{spe_G} \cap S_{spe_{G1}}$ could represent a more accurate identification of the relevant neuron indices.

After obtaining this subset $S$, we ensure that the neuron weights in this part are not modified during the local operation. Then, we sort the remaining neurons based on the absolute value of $\tau_G$ and apply the previously proposed criteria for processing:

$$
\Theta_{\text{global+local}} = \begin{cases} \Theta_{\text{global}} + \tau_G, & \text{neurons in } \overline{S} \text{ and} \\ & \text{within addition range} \\ \Theta_{\text{global}}, & \text{neurons in } S \\ \Theta_{\text{global}} - \tau_G, & \text{neurons in } \overline{S} \text{ and} \\ & \text{within subtraction range} \end{cases} \tag{5}
$$

Finally, as observed from the results in Table 2, after excluding the parts most critical to the specific domain capabilities, the model performs better in the specific domain with smaller performance changes after merging. In short, we further add local operations to the global operation model (the combination of the two is also called the combined model in the following text) to ensure that the specific domain capabilities of the operated model do not drop significantly; furthermore, in local operation, different perturbation methods are selected for different neurons. This perturbation method further improves the overall sensitivity of the model (see the comparison between the green line and the blue line in Figure 6), thereby achieving a better unmergeable effect.

## 3 VERIFICATION EXPERIMENT

We evaluate our approach on a range of mainstream model-merging methods. Section 3.1 describes the experimental setup. Section 3.2 reports the main results, showing that our technique preserves domain performance while rendering the model unmergeable. Section 3.3 presents ablation studies on key merging hyperparameters, and Section 3.4 examines the method's efficiency and robustness.

### 3.1 SETUP

We use Mistral-7B-v0.1 as our ba se and MetaMath-Mistral-7B as our specific model. Compared to the base model, the specific model has made significant breakthroughs in mathematical capabilities, which we consider as the specific domain capability we aim to protect. For model merging methods, we have selected representative approaches such as TA (Ilharco et al., 2023), TIES (Yadav et al., 2024), and DT (Dare TIES) (Yu et al., 2024). These methods encompass key ideas in model merging, including sampling, renormalization, and the use of task vectors. All experiments were conducted on NVIDIA A100 GPUs with 40GB of memory.

**Details of the global operation.** In the global operation, we utilize the models BioMistral-7B and Trendyol-LLM-7b-base-v1.0, which have weaker mathematical capabilities, as the sources of our universal perturbation. Specifically, we select BioMistral-7B as $\theta_1$ and Trendyol-LLM-7b-base-v1.0 as $\theta_2$. After obtaining their respective task vectors for the specific model, we combine them using $\lambda_1 = 0.3$ and $\lambda_2 = 0.05$ to determine the direction of the universal perturbation. Through this operation and applying Eq 2, we obtain the global model.

**Details of the local operation.** Based on the global model, we first obtain the task vector $\tau_G$ by comparing the global model with the base model. Then, we add a small perturbation to the global model to obtain $G_1$. This perturbation is simply achieved by performing a basic TA model merging between the global model and BioMistral-7B. Next, we identify the top 20% of neurons with the highest relevance scores for the GSM8K task using WANDA scoring Sun et al. (2024) from both models to obtain $S_{spe_G}$ and $S_{spe_{G1}}$. Then we take the intersection of the neuron indices from both models, which we denote as the set $S$. After that, we sort the neurons not included in $S$ based on the absolute values of $\tau_G$ in descending order. Based on the sorted order, we treat 2% of the neurons in the sorted list as a unit, and sequentially add and subtract $\tau_G$ to the $\theta_{global}$ for these neurons to create our combined model.

We note that **the combined operated model refers to the model obtained by further applying the local operation to the global model**. The local operation can also be directly applied to the specific model. For clarity, we refer to **the model where the local operation is directly applied to the specific model as the local processing model**.

## 3.2 MAIN RESULTS

In this subsection we aim to empirically demonstrate four key properties of our approach: (i) its effectiveness against multiple model-merging techniques and multiple models that attempt to acquire domain-specific capabilities through merging; (ii) its robustness under adaptive attacks; and (iii) its consistent efficacy across diverse LLM families and a variety of downstream tasks.

**Our method remains effective across diverse model-merging techniques and models seeking to acquire domain-specific capabilities.** To demonstrate the strong generalization capability of our method, we first select the GSM8K dataset as our benchmark for mathematical ability. GSM8K consists of 8,000 crowd-sourced math problems designed for training large language models (LLMs). It includes problems like addition, subtraction, multiplication, division, and basic algebra, with each problem paired with a solution and reasoning. Below, we present the experimental results of the model on GSM8K with five-shot learning.

Before presenting the detailed results, we first introduce a more intuitive metric to assess the effectiveness of our unmergeable method. Specifically, we measure the drop ratio, which quantifies how much of the performance gain achieved by the specific model (compared to the base model) is eliminated by our processing:

$$\text{DropRatio} = \frac{Acc_{\text{spec}} - Acc_{\text{proc}}}{Acc_{\text{spec}} - Acc_{\text{base}}}. \tag{6}$$

where $Acc_{\text{spec}}$, $Acc_{\text{proc}}$ $Acc_{\text{base}}$ are the accuracies with the specific model, our processed model, and the base model, respectively. Notably, the drop ratio larger than one shows that our unmergeable method pushes performance below that merged with the base model.

A key characteristic of an unmergeable model is its ability to maintain comparable specific domain capabilities to a specific model before model merging. As indicated in the "Before Merging" column of Table 3, the specific model attains 71.87 accuracy on GSM8K. Both local and global operations only slightly reduce accuracy and their combination leads to only a 6.8 percent drop, yielding 65.07 accuracy on GSM8K—still well above the baseline accuracy of 38. This highlights that our method can effectively preserve the specific domain capability of the original model, crucial for the concept of unmergeable.

Table 3: Results of different processed models after various model merging methods. Values in parentheses indicate the Drop Ratio, with higher values denoting stronger unmergeability; a value greater than 1 indicates performance even lower than the base model, fully achieving unmerging. All results are averaged over three runs.

| Model | Before Merging | BioMistral-7B | | | Mistral-instruct-v0.2 | | |
|---|---|---|---|---|---|---|---|
| | | TA | TIES | DT | TA | TIES | DT |
| Base model | 38.00 | 34.67 | 19.18 | 32.75 | 43.44 | 43.75 | 40.26 |
| Specific model | 71.87 | 54.06 | 40.79 | 50.34 | 59.00 | 57.47 | 56.48 |
| Local operation | 68.69 | 51.08 (0.15↓) | 37.76 (0.14↓) | 46.85 (0.20↓) | 56.79 (0.14↓) | 53.42 (0.30↓) | 52.28 (0.26↓) |
| Global operation | 67.32 | 34.12 (1.03↓) | 20.02 (0.96↓) | 33.89 (0.94↓) | 53.15 (0.38↓) | 52.08 (0.39↓) | 51.63 (0.30↓) |
| Combined method | 65.07 | 32.74 (1.10↓) | 18.95 (1.01↓) | 31.92 (1.05↓) | 50.46 (0.55↓) | 50.02 (0.54↓) | 48.18 (0.51↓) |

We follow the recommendations of Ilharco et al. (2023) and Yadav et al. (2024), setting both the weight and density to 0.4. To assess the generalizability, we choose Mistral-instruct-v0.2 and BioMistral-7B to acquire mathematical capabilities through merging our processed models. As shown in Table 3, we observe that regardless of the merging method and different models, after applying our proposed combined method, the specific domain capability of the models exhibits a substantial drop than the model merged with the specific model (in the row of "combined", every case shows the drop ratio greater than 50%, with some reductions approaching 100%).

Taking BioMistral-7B as the model that seeks to acquire math capability, we conduct the following analysis: For the TA method, applying only the global operation reduces performance to the baseline level, and adding the local operation leads to further degradation. In the TIES setting, the global operation alone causes a 96.1% loss in specific capability, and the addition of the local operation drops performance below the baseline. Similarly, for DT, global operation leads to a 93.5% loss, and combining both operations further worsens performance beyond the baseline.

Notably, for the original model's specific domain capabilities, even with the combined method, we lead to only a 6.8 percent drop on the GSM8K accuracy. However, after model merging, the rate of decline in mathematical performance is several times higher than that before merging, and in some cases the capability is almost completely lost, which demonstrates the unmergeable effect we emphasize.

**Our method remains robust after merging and can not be easily reversed by adaptive attacks.** To demonstrate the robustness of our approach, we conduct an experiment in which, after model merging, both the specific model and our combined model undergo a single-epoch LoRA-based supervised fine-tuning (SFT) on 100 GSM8K samples, employing a learning rate of 1e-5 and a LoRA rank of 16.

As shown in Table 4, SFT brings no gain to the model merged with the specific model due to the already strong baseline. Furthermore, SFT on the model merged with our processed model further degrades performance, indicating that our method pushes the model into a highly sensitive region that resists recovery through fine-tuning. In other words, our method can not be easily reversed by adaptive attacks such as fine-tuning.

Table 4: GSM8K performance before and after SFT following model merging. All results are averaged over three runs.

| GSM8K | Specific | Combined |
|---|---|---|
| Before SFT | 54.06 | 32.74 |
| After SFT | 52.70 | 1.90 |

Table 5: Drop Ratio changes on MEDQA under different operations. All results are averaged over three runs.

| Drop Ratio | Global | Local | Combined |
|---|---|---|---|
| Before Merging | 18.6 | 5.2 | 21.6 |
| After Merging | 41.6 | 11.3 | 51.2 |

**Our method demonstrates robust generalization across diverse families of LLMs and different tasks.** To further assess generality, we extend our method to the Llama family. Using Meta-Llama-3-8B-Instruct as the baseline and Bio-Medical-Llama-3-8B as the specific medical model. We then apply our unmergeability transformation to ensure that no other model can inherit its medical expertise via model merging. Specifically, we leverage the dad1909/CyberSentinel model, which exhibits comparatively weak performance in the medical domain, to acquire medical capabilities. For the UP selection, we choose lightblue/suzume-llama-3-8B-japanese and meta-llama/Meta-Llama-3-8B-Instruct, while the local operation configuration remains identical to our previous setup.

Using the MEDQA dataset, which assesses domain-specific medical question-answering proficiency, in Table 5 we show that our combined method yields a drop ratio of 21.6, with the accuracy decreasing by merely 3.9 percent on MEDQA. In contrast, after TA merging, the drop ratio increases to 51.2, which is 2.4 times higher than the ratio before merging. These results demonstrate that our method remains effective beyond mathematical tasks, highlighting its robustness and generalizability across both model families and task domains.

## 3.3 ABLATION STUDY

**Influence on Different Merging Parameters** To thoroughly validate our method, we also test multiple weights for each merging method. Here, the weight refers to the $\lambda$ parameter in different model merging methods. Specifically, we experiment with three parameter sets: 0.3, 0.5, and 0.7. Each point in the plot represents the proportion of domain-specific capabilities lost by the processed model compared to the specific model.

Figures 5 demonstrate that under different weights, both local operation and global operation can achieve a certain degree of the unmergeable effect. More importantly, when these two operations are combined, the processed combined model, after merging with the TA, TIES, and DT methods,

further outperforms the individual operations. Additionally, when the weight is less than or equal to 0.4 (for TA) or 0.5 (for TIES and DT), the performance of the combined model falls below the base model, thus achieving the maximum unmergeable effect. These results highlight its robustness and efficiency.

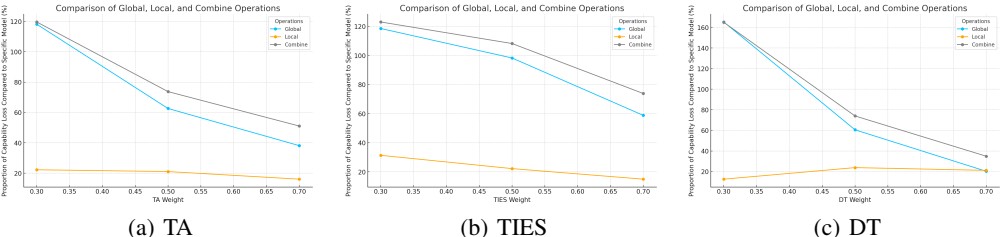

| (a) TA | (b) TIES | (c) DT |

Figure 5: The domain-specific capabilities lost by the processed model in different scenarios compared to the specific model after model merging.

**Sensitivity Visualization**    We then discuss the sensitivity analysis for the models after various operations. The x-axis in Figure 6 represents the Gaussian noise variance. The visualization results indicate that as the Gaussian noise variance increases, the performance of processed models declines more noticeably. Notably, the sensitivity of local and global models exceeds that of the specific model, with the combined model exhibiting the highest sensitivity. This indicates that our methods achieve the unmergeable effect by pushing the model weights into a more sensitive region and the sensitivity analysis align the unmergeable performance across a wide range of settings.

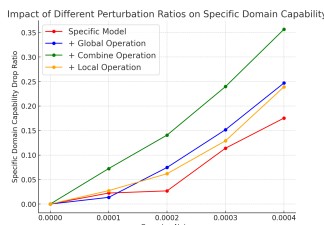

Figure 6: Sensitivity among different operations.

### 3.4 FURTHER DISCUSSION

To further substantiate the robustness of our proposal, we verify that models processed by our method maintain the specific model's performance on other tasks, and we demonstrate that the unmergeable method is not achieved by slightly weakening the model's original domain-specific capabilities. Detailed results are presented in Section A.3 due to space limitations.

Specifically, as shown in Table 6, we evaluate the specific model, local processed model, global processed model, and their combination on the MMLU benchmark. Across diverse tasks such as Humanities and STEM, all processed models achieve performance comparable to that of the specific model. These findings confirm the robustness of our approach and demonstrate that our operations do not impair performance on tasks outside the target domain.

Furthermore, Table 7 shows that our combined processed model retains substantially greater domain-specific capability than the two comparison models. Crucially, regardless of the merging technique, BioMistral-7B is unable to extract more mathematical ability from our processed model than it can from the two comparison models. This indicates that our approach does not rely on the slight degradation of the model's original domain-specific capabilities to achieve the unmergeable effect.

## 4 CONCLUSION

Our paper introduces a new perspective for safeguarding against unauthorized model merging. The proposed unmergeable strategy can be applied before a model is misused thereby addressing limitations such as undetected unauthorized usage or fingerprint failures. From the perspective of neuron-sensitive weight regions, our method prevents unauthorized users from acquiring domain-specific capabilities through model merging. We provide a formal theoretical proof of this protective property, along with extensive experimental evidence demonstrating the effectiveness, robustness, and strong generalization of our approach.

ETHICS STATEMENT

This work makes use of publicly available datasets and models. No private or sensitive data are involved, and no harmful content is included. Therefore, we believe this paper does not raise any ethical concerns.

REPRODUCIBILITY STATEMENT

We provide detailed descriptions of the training and evaluation procedures used in our experiments. The code will be released upon the publication of this paper.

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

## A  APPENDIX

### A.1  RELATED WORK

**Model Merging** offers the advantage of combining multiple task-specific models into a single model without requiring additional training, addressing the limitations of individual models that can only perform a single task and do not benefit from each other (Goddard et al., 2024; Jin et al., 2022; Lu et al., 2024; Davari & Belilovsky, 2025; Ilharco et al., 2023; Yadav et al., 2024; Yu et al., 2024). Linear Mode Connectivity has become a mainstream approach for model merging, using linear interpolation to combine neural network checkpoints while requiring identical architectures and initializations across the models. The simplest merging method is weight averaging, which relies on linear mode connectivity and serves as the foundation for other methods. Ilharco et al. (2023) introduces task vectors, demonstrating the effectiveness of arithmetic operations on the differences between fine-tuned models and a baseline model. Methods such as TIES merging (Yadav et al., 2024) and DARE (Yu et al., 2024) introduce sparsification and combination techniques for task vectors, allowing the merging of more models without performance degradation. These methods do not require training data or fine-tuning after merging, thus making them more practically valuable. This paper aims to propose a method that prevents unauthorized users from exploiting the model owner's model through the aforementioned model merging technique. A detailed explanation will be provided in the following sections.

**Model intellectual property (IP) protection** has received increasing attention in recent years (Zhang et al., 2021; He et al., 2022b;a; Yang et al., 2021; Ong et al., 2021; Shao et al., 2024). As machine learning models are widely applied , they face risks of unauthorized use or replication as valuable assets. Watermarking and fingerprinting are two commonly used, closely related but slightly different methods for model IP protection. The fingerprints of large language models can be mainly classified into two types. One type involves adding fingerprints during the model's training or fine-tuning process, which includes embedding a backdoor trigger into the model. With these triggers, it becomes possible to verify whether a suspicious model is derived from the victim model (Zhang et al., 2018; Li et al., 2023; Peng et al., 2023; Kirchenbauer et al., 2023). In fact, the watermarking approach we refer to is essentially a method where researchers inject fingerprints. Therefore, it can be seen that watermarking, in a certain sense, is a part of the fingerprinting method. The other type utilizes the inherent characteristics of the model as fingerprints. These fingerprints are based on the model's intrinsic properties and cannot be removed (McGovern et al., 2024; Xu et al., 2024; Refael et al., 2024). The difference between this paper and the aforementioned fingerprint methods is that fingerprints serve as a post-protection mechanism for models, protecting intellectual property after the model may have already been exploited by unauthorized users or malicious actors. In contrast, the concept of unmergeable introduced in this paper, along with the proposed solution, represents a form of preemptive protection—where the model is processed before being released by the model owner. Although PaRaMS (Zhou et al., 2024) also aims to block parameter merging, it depends on reversible permutations and scaling matrices that serve as a secret key; if this key is lost or leaked, the protection vanishes. Our scheme is irreversible and thus avoids this single point of failure. PaRaMS was developed for medium scale vision and text models, and applying it to LLMs with thousands of attention heads would require managing thousands of matrix pairs, which greatly increases engineering cost and attack surface. We design our method specifically for LLMs so it remains scalable and free of key-management issues.

### A.2  THEORETICAL PROOF

*Proof.* The proof for the unmergeable goal (see Eq. 1) in the main text, Section 2.2, proceeds as follows:

According to Taylor expansion (ignoring higher-order infinitesimals), we have

$$L(w + \delta + \mu) - L(w + \delta) = \nabla L(w + \delta)\mu \tag{7}$$
$$L(w + \mu) - L(w) = \nabla L(w)\mu \tag{8}$$

Since the model is pushed to the more sensitive region, we have $\nabla L(w) < \nabla L(w + \delta)$, which proves that model merging causes a larger performance drop for our processed model, i.e., our processing achieves the unmergeable effect.

$\square$

## A.3 ADDITIONAL RESULTS

**Our unmergeable model matches the performance of the specific model on various tasks across different domains except specific domain.** To validate this, we conduct a comprehensive evaluation of the performance differences between our proposed unmergeable model and the specific model across a variety of tasks on the MMLU benchmark.

Table 6: Performance on MMLU tasks under different model. All results are averaged over three runs.

| Task | Specific | Global | Local | Combined |
|------|----------|--------|-------|----------|
| Humanities | 53.11 | 52.99 | 52.94 | 52.43 |
| Sciences | 68.48 | 70.82 | 68.38 | 69.94 |
| STEM | 47.76 | 49.48 | 48.14 | 49.48 |
| Other | 66.91 | 67.43 | 66.62 | 67.52 |
| Average | 58.33 | 59.30 | 58.28 | 58.94 |

As illustrated in Table 6, our unmergeable model even surpasses the specific model in average MMLU performance, indicating that our processing not only prevents the transfer of domain-specific capabilities via model merging but also imposes no detrimental effects on performance across tasks in other domains.

**Our model does not achieve the unmergeable effect by reducing the specific domain capabilities of the model.** In Table 7, we show that the models SciMistral-V1 and Mistral-instruct-v0.2 both possess certain mathematical abilities. Accordingly, we merge BioMistral-7B with (i) our protected combined model, (ii) SciMistral-V1, and (iii) Mistral-Instruct-v0.2 to acquire their mathematical capabilities. As shown in Table 7, we can see that the specific domain capability of our combined model processed is significantly higher than the other two models. However, it is important to note that, regardless of the model merging method used, BioMistral-7B cannot acquire more mathematical ability from our processed model than from the other two models. From this perspective, our method has indeed protected our model, achieving the unmergeable effect.

Table 7: Results under different models which BioMistral-7B try to acquire mathematical capability. All results are averaged over three runs.

| Model | Original | TA | DT | TIES |
|-------|----------|-----|-----|------|
| Combined Model | 65.07 | 32.74 | 31.92 | 18.95 |
| SciMistral-V1 | 56.71 | 44.05 | 42.76 | 27.29 |
| Mistral-instruct-v0.2 | 42.10 | 43.82 | 38.44 | 34.95 |

Furthermore, Table 7 demonstrates that our method does not achieve the unmergeable effect by reducing the specific domain capabilities of the model (the subtle difference in specific domain capabilities between the specific model and the combined model). Instead, it achieves this effect by pushing the model parameters into a more sensitive region as we analyzed before.

## A.4 USAGE OF LLM

We commit to using LLMs for text polishing based on prompts. All polished text are double-checked by authors to ensure accuracy, avoid over-claims, and prevent confusion.

