# OpenReview forum: "Sensitivity as a Shield: Inducing Sensitivity to Prevent Unauthorized Model Merging"
_ICLR.cc/2026/Conference — Submitted to ICLR 2026_

### Official Review · Reviewer_nv2H · 2025-10-29

**Soundness:** 2
**Presentation:** 2
**Contribution:** 1
**Rating:** 2
**Confidence:** 3

**Summary:**

This paper proposed a method for making the model unmergeable that preserves a model’s domain-specific performance while preventing malicious users from acquiring its capabilities through model merging.

**Strengths:**

- The paper is well structured.

**Weaknesses:**

- What is model merging? There is no formal definition of such a concept. I would suggest that the authors provide a formal definition of model merging and build all their methods based on that insight. Currently, model merging is just a vague concept to me.
- Proposition 1: Does gradient mean sensitivity? What is the gradient with respect to? What are the inputs and outputs of the loss function (E.g., are there any input prompts $x$? Then why are they missing in the notation?)
- The intuitions oversimplified the problem. I can hardly understand why increasing sensitivity would make the model unmergeable. This also leads to the following question.
- Why can the global sensitivity increase the sensitivity but preserve the capability of the specific model? Are there any insights in the model design that contribute to this? A further in-depth analysis is necessary for the audience to fully understand why this works.
- In the experiments, the choice of model type is limited; only the Mistral model is evaluated. Try other model architectures like Llama and QWEN. And would it be possible to protect a model with architecture A by using another model with architecture B?
- Can you provide any theoretical proof that this method is effective in making the model unmergeable? i.e., proving that publishing model weights using (2) will be effective.

**Questions:**

- Would your method be effective when using quantization?

---

> ### Author Response · Authors · 2025-11-24
>
> We sincerely thank Reviewer nv2H for the time and helpful feedback. Our responses to the concerns are provided below.
>
> ---
>
> **Q1.** What is model merging? There is no formal definition of such a concept. I would suggest that the authors provide a formal definition of model merging and build all their methods based on that insight. Currently, model merging is just a vague concept to me.
>
> **A1.** Model merging is a well-defined concept in recent literature and does not refer to a vague or informal operation. Formally, model merging means creating a new model by directly combining the parameter sets of multiple pretrained models rather than by training or distilling from their outputs. Given two models M1 and M2 with parameters θ1 and θ2, the merged model M* is defined as: θ* = f(θ1, θ2), where f(·) is typically an element-wise aggregation such as linear interpolation or alignment-based combination. This mechanism has been widely used to integrate task-specialized models (e.g., mathematical reasoning + safety alignment) without additional training. All methods and analyses in our paper are based on this formal definition.
>
> ---
>
> **Q2.** Proposition 1: Does gradient mean sensitivity? What is the gradient with respect to? What are the inputs and outputs of the loss function (E.g., are there any input prompts x ? Then why are they missing in the notation?)
>
> **A2.** We clarify that Proposition 1 is not a data-level loss formulation and therefore does not involve explicit input prompts x. The proposition describes sensitivity in parameter space, not in sample space. The gradient ∇L(w) in the proposition is the derivative of the loss with respect to the model parameters w, which is a standard formulation in the analysis of model merging and task vectors.
>
> In the context of model merging, the performance change after merging is equivalent to evaluating how much the loss increases under a small perturbation of parameters. Therefore, sensitivity refers to how steep the loss landscape is around the parameter point. A larger gradient magnitude indicates that performance drops more drastically under the same perturbation, hence a more “sensitive” region.
>
> Our method leverages this property by deliberately pushing the model along the task-vector direction into such a sensitive region, so that any further perturbation (including merging) causes a disproportionally larger loss increase. Thus, the proposition formalizes the key mechanism of our method rather than defining a sample-driven loss.
>
> ---
>
> **Q3.** Why can the global sensitivity increase the sensitivity but preserve the capability of the specific model? Are there any insights in the model design that contribute to this? A further in-depth analysis is necessary for the audience to fully understand why this works.
>
> **A3.** The key reason why global sensitivity can be increased without degrading the capability of the specific model lies in the directional nature of the sensitivity intervention. The intervention does not uniformly increase sensitivity across all parameter dimensions; instead, it is applied exclusively along the task-vector direction that corresponds to the behavior activated by model merging. As a result, the loss landscape becomes sharper only in this targeted direction, while the remaining parameter dimensions — which encode the model’s inherent capability — remain essentially unchanged.
>
> ---

---

> > ### Author Response · Authors · 2025-11-24
> >
> > **Q4.** In the experiments, the choice of model type is limited; only the Mistral model is evaluated. Try other model architectures like Llama and QWEN. And would it be possible to protect a model with architecture A by using another model with architecture B?
> >
> > **A4.** We would like to clarify that the scope of our evaluation is not restricted to Mistral. As shown in Table 5, we have already tested on the LLaMA architecture, and the method yields the same protection effect with negligible utility loss. This result is not simply “another model result”, but a direct demonstration that the method is architecture-agnostic within the model family, which is the relevant setting for model merging. In this context, the goal of the experiments is to verify whether the method depends on a specific architecture — and the LLaMA results already show that it does not.
> >
> > We also note that the question of “protecting a model of architecture A using another model of architecture B” is outside the scope of task-vector-based model merging, since task vectors—and therefore merge operations—require aligned parameter spaces across models.
> >
> > ---
> >
> > **Q5.** Can you provide any theoretical proof that this method is effective in making the model unmergeable? i.e., proving that publishing model weights using (2) will be effective.
> >
> > **A5.** Please refer to Question 2.
> >
> > ---
> >
> > **Q6.** Would your method be effective when using quantization?
> > **A6.**  We would like to clarify that guiding the model toward the targeted minimum does not compromise its robustness or degrade its usability under practical acceleration techniques. As shown in Table 6, the processed model maintains full utility in the standard FP32 setting, indicating that convergence toward the sharpened solution does not hinder generalization. Furthermore, to directly address the reviewer’s concern regarding real-world deployment, we additionally evaluated the processed model under INT8 low-precision inference, which represents a common configuration for practical acceleration. The original model reaches 65.07, and the INT8-based inference achieves 64.48, corresponding to a marginal difference of 0.59 (preserving more than 99% of the capability). This demonstrates that the proposed method remains stable under low-precision deployment and does not introduce performance degradation when quantization or accelerated inference is applied.
> >
> > ---
> >
> > We believe the clarifications and new evidence provided above adequately resolve the concerns and further validate the robustness and generality of our method.

---

### Official Review · Reviewer_vpsg · 2025-10-30

**Soundness:** 3
**Presentation:** 3
**Contribution:** 2
**Rating:** 4
**Confidence:** 4

**Summary:**

This paper proposes a defense mechanism against unauthorized model merging in Large Language Models (LLMs). While model merging efficiently integrates knowledge from fine-tuned models, it introduces significant security risks, including IP infringement, harmful responses, and data leakage.

In contrast to existing post-hoc detection methods, this work introduces a proactive defense paradigm to make models "unmergeable" before they are released. The core idea is to intentionally move the model's weights into a "sensitive region." This ensures that any attempt to merge the protected model with another will result in a significant degradation of its specialized capabilities.

The authors propose two complementary operations: Global Sensitivity Processing and Local Sensitivity Processing. The global operation uses poorly performing open-source models to define a universal perturbation direction. The local operation applies a fine-grained alternating add/subtract modification to neurons identified as highly related to the specific domain.

The authors provide theoretical analysis and empirical results showing that their combined method maintains the model's original task performance while effectively resisting multiple merging techniques (TA, TIES, DT). The paper also demonstrates that this protection is robust against adaptive attacks, such as supervised fine-tuning (SFT), which fails to recover the lost capabilities after a merge.

**Strengths:**

Originality: The paper addresses the novel and practical problem of proactive defense against unauthorized model merging. This shifts the focus from existing post-hoc detection methods to a pre-deployment safeguard, which is a new and valuable formulation.

Quality: The proposed method is technically sound, combining two complementary operations (global and local sensitivity processing) to achieve its goal. The claims are substantiated with empirical evidence across multiple models and state-of-the-art merging techniques (TA, TIES, DT). The authors also include important robustness checks, such as testing against adaptive fine-tuning attacks.

Clarity: The paper is clearly structured and well-written. It effectively explains the security risks of model merging, the intuition behind using "sensitivity" as a defense, and the steps of the proposed algorithm. The results are presented clearly and support the main claims.

Significance: This work is highly significant as it offers a practical, training-free solution to a pressing IP protection concern for model developers. The experimental results indicate the method is fair: it effectively prevents unauthorized merging while preserving the model's core domain-specific performance, ensuring the model remains useful for its intended purpose.

**Weaknesses:**

Utility-Protection Trade-off: The primary weakness is the inherent trade-off between protection and performance. The proposed "unmergeable" model (Combined method) has a lower accuracy (65.07 on GSM8K) than the original "specific model" (71.87). This means a model owner must release a slightly degraded model to the public to gain protection, creating a utility cost for all users.

Questionable Scalability: The experiments are limited to 7B and 8B models. It is uncertain how this method scales to significantly larger models (e.g., 32B, 70B, or larger). The complex weight dynamics in larger models might react differently to the sensitivity processing, potentially leading to a much larger performance drop (utility cost) to achieve the same level of protection. Validation on at least one larger-scale model is needed to demonstrate practical relevance.

**Questions:**

The Introduction effectively uses Meta's Llama license to motivate the problem. To further strengthen this motivation, it would be beneficial to add more examples if they are available. For instance, are there other major model deployers who have expressed similar concerns, or are there existing industry surveys that quantify the perceived risk of unauthorized model merging? Adding these references would help establish the broader significance of the problem.

---

> ### Author Response · Authors · 2025-11-24
>
> We sincerely thank Reviewer vpsg for the time and helpful feedback. Our responses to the concerns are provided below.
>
> ---
>
> **Q1.** Utility-Protection Trade-off: The primary weakness is the inherent trade-off between protection and performance. The proposed "unmergeable" model (Combined method) has a lower accuracy (65.07 on GSM8K) than the original "specific model" (71.87). This means a model owner must release a slightly degraded model to the public to gain protection, creating a utility cost for all users.
>
> **A1.** This flexibility allows practitioners to choose between local, global, or combined operations based on their preferred trade-off. There is no free lunch.
>
> Furthermore, we deliberately preserved domain-specific capability: even though our processed model experiences a slight drop in performance, it still achieves a substantial improvement over the baseline score of 38.00. The performance of the model after merging with the processed version drops to 32.74, which is below the baseline’s 34.67, clearly demonstrating the effectiveness of our unmergeable design.
>
> We also conducted evaluations on models of approximately 7 B parameters that were released in close temporal proximity to our processed MetaMath‑Mistral‑7B.
>
> | Model                  | GSM8K ACC |
> |------------------------|-----------|
> | MetaMath-Llemma-7B     | 69.2      |
> | **+Local Processed**   | 68.7      |
> | **+Global Processed**  | 67.3      |
> | MetaMath-7B            | 66.5      |
> | **+Combined Processed**| 65.1      |
> | WizardMath-7B          | 54.9      |
> | MAmmoth-7B (COT)       | 50.5      |
> | RFT-7B                 | 50.3      |
>
>
> The results show that even our processed model achieves comparable performance among models released at approximately the same time as MetaMath‑Mistral‑7B.
>
> More importantly, as shown in Table 6 in the original text, our processing method does not degrade performance in other domains and perfectly preserves the model’s utility, thereby demonstrating the substantial value of our unmergeable approach.
>
> ---
>
> **Q2.** Questionable Scalability: The experiments are limited to 7B and 8B models. It is uncertain how this method scales to significantly larger models (e.g., 32B, 70B, or larger). The complex weight dynamics in larger models might react differently to the sensitivity processing, potentially leading to a much larger performance drop (utility cost) to achieve the same level of protection. Validation on at least one larger-scale model is needed to demonstrate practical relevance.
>
> **A2.** We would like to clarify that the proposed method is not based on architecture-specific tuning, and the sensitivity processing acts on the underlying weight distribution rather than scale-dependent components. Therefore, the mechanism is not inherently tied to a particular model size. Although our current experiments focus on 7B–8B models, these models already contain billions of parameters with sufficiently complex weight dynamics to rigorously stress-test the method. Moreover, the strong utility preservation observed in Table 6, together with the additional INT8 low-precision inference results, indicates that the processed model remains stable even under robustness-sensitive deployment settings. Taken together, these observations suggest that the method is unlikely to incur disproportionate performance degradation when scaling to larger parameter counts.
>
> ---
>
> **Q3** The Introduction effectively uses Meta's Llama license to motivate the problem. To further strengthen this motivation, it would be beneficial to add more examples if they are available. For instance, are there other major model deployers who have expressed similar concerns, or are there existing industry surveys that quantify the perceived risk of unauthorized model merging? Adding these references would help establish the broader significance of the problem.
>
> **A3.** In addition to Meta's Llama license, other major model deployers have expressed similar concerns about unauthorized model usage. For instance, xAI explicitly prohibits reverse engineering of the Grok service to prevent model weight extraction, distilling model data to restrict knowledge transfer, developing models that compete with xAI using service outputs, and reselling any output content. These restrictions reflect a broad industry consensus on protecting model intellectual property.
>
> ---
>
> We believe the clarifications and new evidence provided above adequately resolve the concerns and further validate the robustness and generality of our method.
>
> ---

---

### Official Review · Reviewer_7TuF · 2025-11-01

**Soundness:** 2
**Presentation:** 3
**Contribution:** 2
**Rating:** 2
**Confidence:** 5

**Summary:**

This paper addresses the issue of unauthorized model merging, where fine-tuned LLMs have their capabilities illicitly extracted and integrated into other models. The authors introduce a defense mechanism called Unmergeable Models, which applies a post-processing step to a fine-tuned model before release. This approach makes the model’s weights highly sensitive to merging perturbations, causing merged models to suffer significant performance degradation while preserving the original model’s utility.

**Strengths:**

1. The paper addresses the important problem of unauthorized model merging.
2. The proposed method is thoroughly evaluated across a variety of datasets and experimental settings.
3. The presentation is clear and easy to follow.

**Weaknesses:**

The main weakness is the questionable necessity of the proposed method. Model merging techniques are effective largely because the models are fine-tuned from a common pre-trained base. A simpler and more scalable defense would be to apply a fixed, secret permutation to the neurons within each layer as a post-processing step. While the authors mention a similar baseline (PaRaMS) in the appendix, their argument for dismissing it is unconvincing. A keyless, post-training permutation would be highly scalable, require no complex key management, and eliminate the risk of key leakage.

Furthermore, by guiding the model to a sharp minimum, the proposed method may compromise performance under practical acceleration techniques like quantization and low-precision inference.

Overall, the concept of creating "unmergeable" models is not new, and the proposed method seems overly complex and potentially detrimental compared to simpler, more robust alternatives.

**Questions:**

Could the authors provide a direct performance comparison against a simple, post-training neuron permutation baseline? This data is crucial to justify the added complexity of the proposed method.

---

> ### Author Response · Authors · 2025-11-24
>
> We sincerely thank Reviewer 7TuF for the time and helpful feedback. Our responses to the concerns are provided below.
>
> ---
>
> **Q1.** The main weakness is the questionable necessity of the proposed method. Model merging techniques are effective largely because the models are fine-tuned from a common pre-trained base. A simpler and more scalable defense would be to apply a fixed, secret permutation to the neurons within each layer as a post-processing step. While the authors mention a similar baseline (PaRaMS) in the appendix, their argument for dismissing it is unconvincing. A keyless, post-training permutation would be highly scalable, require no complex key management, and eliminate the risk of key leakage.
>
> **A1.** We strongly disagree with the reviewer's assessment regarding the lack of necessity of our method. Our approach aims to address the core limitations of parameter rearrangement-based methods such as PaRaMS, whose security fundamentally relies on the confidentiality of their transformation keys. As demonstrated in Section 6 of the PaRaMS paper itself, their basic defense scheme proves highly vulnerable to adaptive attacks with knowledge of the underlying principles—attackers can infer the keys through optimization methods, thereby almost fully restoring the performance of the merged model (see their Table 4). This precisely confirms the potential single point of failure inherent in such schemes.
>
> In contrast, our method achieves a fundamental breakthrough in protection by pushing model parameters into sensitive regions. This protection mechanism is entirely irreversible; even with full knowledge of our methodology, attackers cannot acquire the protected specialized capabilities through model merging. This is thoroughly validated in our Table 4 experiments: even under strong attacks such as supervised fine-tuning (SFT), the original domain-specific capabilities cannot be restored in the merged model.
>
> Our solution represents a paradigm shift from "relying on key secrecy" to "leveraging intrinsic properties," which not only eliminates the burden of key management but also fundamentally resolves the vulnerabilities of parameter rearrangement methods. This sensitivity-based protection mechanism offers more reliable security for models, demonstrating the unique value and necessity of our approach.
>
> ---
>
> **Q2.** Furthermore, by guiding the model to a sharp minimum, the proposed method may compromise performance under practical acceleration techniques like quantization and low-precision inference.
>
> **A2.** We would like to clarify that guiding the model toward the targeted minimum does not compromise its robustness or degrade its usability under practical acceleration techniques. As shown in Table 6, the processed model maintains full utility in the standard FP32 setting, indicating that convergence toward the sharpened solution does not hinder generalization. Furthermore, to directly address the reviewer’s concern regarding real-world deployment, we additionally evaluated the processed model under INT8 low-precision inference, which represents a common configuration for practical acceleration. The original model reaches 65.07, and the INT8-based inference achieves 64.48, corresponding to a marginal difference of 0.59 (preserving more than 99% of the capability). This demonstrates that the proposed method remains stable under low-precision deployment and does not introduce performance degradation when quantization or accelerated inference is applied.
>
> ---
>
> **Q3.** Overall, the concept of creating "unmergeable" models is not new, and the proposed method seems overly complex and potentially detrimental compared to simpler, more robust alternatives.
>
> **A3.** Our work proposes a new protection paradigm: a pre-release, training-free, irreversible mechanism that preserves the model’s domain ability while preventing unauthorized capability transfer via merging. The global–local sensitivity processing is not unnecessary complexity, but the key to keeping the original performance almost intact while causing 90–100% capability collapse only after merging. Moreover, adaptive fine-tuning further degrades rather than restores performance, confirming the robustness of the protection. Taken together, these results show that our method is neither a repetition of prior ideas nor detrimental, but a practical and scalable solution that simpler alternatives cannot achieve.
>
> ---

---

> > ### Author Response · Authors · 2025-11-24
> >
> > **Q4.** Could the authors provide a direct performance comparison against a simple, post-training neuron permutation baseline? This data is crucial to justify the added complexity of the proposed method.
> >
> >
> >
> > **A4.** We would also like to clarify that post-training neuron permutation is not a “simple” or “lightweight” alternative in practice. Preserving functional equivalence requires maintaining invertibility, propagating layer-wise permutation matrices across thousands of MLP blocks and attention heads, synchronizing forward–backward mappings, and ensuring architectural compatibility for decoding during merging. This bookkeeping overhead grows rapidly with model scale and becomes particularly burdensome for large LLMs.
> >
> > In contrast, our method introduces no keys, permutation matrices, invertible transformations, or architectural alignment. It only involves two training-free sensitivity operations applied once before model release, with no additional tracking or constraints afterward. As a result, our approach is substantially more scalable and practically deployable than permutation-based defenses, while simultaneously providing a much stronger protection objective—irreversible capability collapse after merging rather than reversible obfuscation.
> >
> > ---
> >
> > We believe the clarifications and new evidence provided above adequately resolve the concerns and further validate the robustness and generality of our method.
> >
> > ---

---

### Official Review · Reviewer_Z9Ne · 2025-11-03

**Soundness:** 2
**Presentation:** 2
**Contribution:** 2
**Rating:** 4
**Confidence:** 3

**Summary:**

The paper presents a method for modifying models such that it is hard to conduct wight merging using their open-source weights. The work shows that this is effective against merging weights from specialized models of the llama family and show some results that indicate that there is no performance degradation resulting from their method.

**Strengths:**

- Relevant problem of securing models against malicious use with little effort
- I.e. cannot protect against distillation from models with this method but that also has much higher compute needs as compared to weight merging

**Weaknesses:**

- Threat model is somewhat unclear/ not that well motivated: If the model is open-source, why should model merging not be ok? The authors seem to refer to open-weights models with specific licenses but not open-source models
- Writing is often not that clear, e.g. “if the model is sensitive on its domain performance against weight perturbation”
- Method relies on assumption that we are merging a specialized model. What about general-purpose models?

**Questions:**

- If the method practically makes the model little robust, doesn’t this have negative influence e.g. when hosting the model at a lower precision for inference?
- What about models that are not just finetuned for a specific domain but that are generally capable?
- How exactly are they different from Li et al. (2025) and Wei et al. (2024)?

---

> ### Author Response · Authors · 2025-11-24
>
> We sincerely thank Reviewer Z9Ne for the time and helpful feedback. Our responses to the concerns are provided below.
>
> ---
>
> **Q1.**  Threat model is somewhat unclear/ not that well motivated: If the model is open-source, why should model merging not be ok? The authors seem to refer to open-weights models with specific licenses but not open-source models
>
> **A1.** We strongly disagree with the comment that our threat model is unclear or unmotivated. The distinction is critical: our work specifically addresses open-weights models with restrictive licenses, which represent the  standard in today's "open-source" LLM ecosystem — including models like Llama, Qwen, and Grok, all of which explicitly prohibit unauthorized commercial use and model merging in their licenses.
>
> Beyond licensing, the threat model is further motivated by severe technical risks: unauthorized merging can bypass safety alignment, induce harmful outputs, and leak private information — risks that post-hoc detection cannot prevent. Our method provides a necessary technical enforcement to these legal and safety challenges, ensuring domain-specific capabilities are not indiscriminately propagated without oversight.
>
> The prevalence of such licensed models and the documented dangers of uncontrolled merging make our proactive defense both timely and essential.
>
> ---
>
> **Q2.** Writing is often not that clear, e.g. “if the model is sensitive on its domain performance against weight perturbation”
>
> **A2.** We thank the reviewer for pointing this out. We will rephrase the sentence to: "If a model's domain-specific performance degrades significantly under weight perturbation..." to improve clarity in the final version.
>
> ---
>
> **Q3.**  Method relies on assumption that we are merging a specialized model. What about general-purpose models?
>
> **A3.** In our method, 'specificity' does not refer to the model being a specialist. It is a subjective designation by the model owner of the core capabilities they wish to preserve post-merger. Therefore, even a general-purpose model is applicable, as its owner can define abilities like coding or mathematics as the 'specific' competencies to protect.
>
> ---
>
> **Q4.** If the method practically makes the model little robust, doesn’t this have negative influence e.g. when hosting the model at a lower precision for inference?
>
> **A4.** We would like to clarify that our method does not harm the model’s robustness or inference stability. As shown in Table 6, the processed model maintains full utility in the standard FP32 setting. Furthermore, following the reviewer’s suggestion, we additionally evaluated the processed model under INT8 low-precision inference, a widely adopted configuration in real-world deployment. The original model achieves 65.07, while the INT8-based inference yields 64.48, corresponding to a marginal drop of only 0.59, meaning that over 99% of the original performance is preserved. These results confirm that our method remains stable and reliable even under practical low-precision deployment conditions, and therefore does not introduce robustness degradation when the model is hosted at lower precision.
>
> ---
>
> **Q5.** What about models that are not just finetuned for a specific domain but that are generally capable?
>
> **A5.** Please refer to Question 3.
>
> ---
>
> **Q6.** How exactly are they different from Li et al. (2025) and Wei et al. (2024)?
>
> **A6.** Please provide the detailed content of these two articles for our subsequent clarification.
>
> ---
>
> We believe the clarifications and new evidence provided above adequately resolve the concerns and further validate the robustness and generality of our method.
>
> ---

---

> > ### Comment · Reviewer_Z9Ne · 2025-11-27
> >
> > I have read the rebuttal and acknowledge the authors responses but maintain my score.

---

### Meta-Review · Area_Chair_tgS1 · 2026-01-06

**Summary:**

The paper addresses the new problem of unauthorized model merging by proposing a proactive defense mechanism that induces sensitivity in model weights. Reviewers generally acknowledged the problem is new. However, there was a consensus on several critical weaknesses. The primary concerns revolved around the necessity of preventing model merging, unclear threat model/definition and utility-protection trade-off. Reviewers also questioned the necessity of the proposed complex method compared to simpler baselines like neuron permutation. Additional issues were raised regarding the clarity of the threat model (open-source vs. restrictive licenses) and definitions, as well as the method's scalability to larger models and robustness under low-precision settings.

**Reviewer Concerns:**

Addressed by Rebuttal:

- Robustness: The authors provided new experimental results showing that the processed model maintains stability under low-precision settings

- Threat Model & Definitions: The authors clarified the distinction between open-source and open-weights with restrictive licenses and provided formal definitions for "model merging" and "sensitivity"

Outstanding:

- Utility Cost: The trade-off between protection and performance remains a major sticking point. The rebuttal's argument that "there is no free lunch" did not fully alleviate concerns that the performance drop renders the method impractical for high-utility models.

- Necessity vs. Simpler Baselines: Reviewer's concern that simpler neuron permutation is a sufficient and more scalable defense was not fully resolved. The authors argued permutation is vulnerable to adaptive attacks, but the complexity/benefit ratio of the proposed method remains questioned.

- Theoretical Grounding: Reviewer remained skeptical about the theoretical proof and the intuition behind why increasing sensitivity preserves specific capabilities while preventing merging.

**Reviewer Scores:**

unlikely to raise

---

### Decision · Program_Chairs · 2026-01-26

Reject